# Comparison of the Efficiency of Selected Disinfectants against Planktonic and Biofilm Populations of *Escherichia coli* and *Staphylococcus aureus*

**DOI:** 10.3390/microorganisms11061593

**Published:** 2023-06-15

**Authors:** Olja Todorić, Lato Pezo, Ljubiša Šarić, Violeta Kolarov, Ana Varga, Ivana Čabarkapa, Sunčica Kocić-Tanackov

**Affiliations:** 1Institute of Food Technology, University of Novi Sad, Bulevar cara Lazara 1, 21000 Novi Sad, Serbia; ljubisa.saric@fins.uns.ac.rs (L.Š.); ana.varga@fins.uns.ac.rs (A.V.); 2Institute of General and Physical Chemistry, University of Belgrade, Studentski trg 12/V, 11000 Belgrade, Serbia; latopezo@yahoo.co.uk; 3Faculty of Medicine, University of Novi Sad, Hajduk Veljkova 3, 21000 Novi Sad, Serbia; violeta.kolarov@mf.uns.ac.rs; 4Faculty of Technology, University of Novi Sad, Bulevar cara Lazara 1, 21000 Novi Sad, Serbia; suncicat@uns.ac.rs

**Keywords:** quaternary ammonium compound, peracetic acid, biofilm, *Escherichia coli*, *Staphylococcus aureus*, *Chromobacterium violaceum*, quorum sensing

## Abstract

The aim of this study is to compare the efficacy of selected food disinfectants on planktonic populations of *Staphylococcus aureus* and *Escherichia coli* and on the same microorganisms (MOs) incorporated in a biofilm. Two disinfectants were used for treatment: peracetic acid-based disinfectant (P) and benzalkonium chloride-based disinfectant (D). Testing of their efficacy on the selected MO populations was performed using a quantitative suspension test. The standard colony counting procedure was used to determine their efficacy on bacterial suspensions in tryptone soy agar (TSA). The germicidal effect (GE) of the disinfectants was determined based on the decimal reduction ratio. For both MOs, 100% GE was achieved at the lowest concentration (0.1%) and after the shortest exposure time (5 min). Biofilm production was confirmed with a crystal violet test on microtitre plates. Both *E. coli* and *S. aureus* showed strong biofilm production at 25 °C with *E. coli* showing significantly higher adherence capacity. Both disinfectants show a significantly weaker GE on 48 h biofilms compared to the GE observed after application of the same concentrations on planktonic cells of the same MOs. Complete destruction of the viable cells of the biofilms was observed after 5 min of exposure to the highest concentration tested (2%) for both disinfectants and MOs tested. The anti-quorum sensing activity (anti-QS) of disinfectants P and D was determined via a qualitative disc diffusion method applied to the biosensor bacterial strain *Chromobacterium violaceum* CV026. The results obtained indicate that the disinfectants studied have no anti-QS effect. The inhibition zones around the disc therefore only represent their antimicrobial effect.

## 1. Introduction

Nowadays, biofilms are one of the most studied contaminants in the food industry, as they can pose a threat to food safety. Numerous studies have confirmed that biofilms formed by bacteria on various surfaces in the food industry are a long-term source of food contamination not only with bacteria causing spoilage but also with foodborne pathogens, such as *Salmonella* spp., *Campylobacter* spp., *Escherichia coli* and *Listeria monocytogenes* [1,2]. Foodborne pathogens in food processing plants can exist for several months or even years. These strains are referred to as “house strains”, and their existence is thought to be enabled by their ability to form biofilms [3]. Previous studies confirmed the ability of foodborne pathogens to adhere to and form biofilms on various types of materials commonly used in the food industry [2,4,5,6,7,8,9,10]. Recent studies have shown that the cleaning process can remove more than 90% of surface-associated MOs. However, they cannot be completely destroyed by the cleaning and disinfection process, indicating the presence of residual bacteria that are normally trapped in a biofilm [11]. Previous reports suggest that the presence of bacteria in biofilms contributes to the acquisition of tolerance to cleaning and disinfection agents [12,13,14,15]. One of the regulatory mechanisms by which bacteria respond to external environmental stresses through the expression of a large number of genes is quorum sensing (QS), which is generally defined as a population-controlled bacterial communication process. QS regulates numerous important cellular functions in both Gram-positive and Gram-negative bacteria, including metabolism, protein synthesis, virulence factor expression, antibiotic resistance, biofilm formation, biofilm maintenance and spread and entry into the stationary phase [16].

Exposure of bacteria to sublethal concentrations of disinfectants can increase their tolerance to antibiotics, an effect that inevitably has implications when the associated public health risks are considered [17,18]. However, the use of disinfectants, an important step in preventing the transmission of bacteria in industrial environments, should be taken into account, as recommended concentrations of disinfectants usually refer to the planktonic populations of MOs. Nevertheless, the aforementioned methods are insufficient for the evaluation of antimicrobial efficacy against cells entrapped in biofilms. It has been estimated that biofilms can tolerate antimicrobial agents (disinfectants, surfactants) at concentrations 10–1000 times higher than those required to inactivate, genetically equivalent planktonic bacteria [19]. Almost all industrially approved antimicrobials are less active against bacteria trapped in a biofilm than against planktonic cells [20].

Our previous research revealed the presence of residual bacteria after cleaning and disinfection. These findings point out that the washing and disinfection procedures are not effective enough to eradicate microorganisms in real industrial environments. The most frequently isolated microorganisms were *E. coli* (10), *S. aureus* (5) and *Pseudomonas* spp. (2). Based on the findings obtained for this research study, *E. coli* was chosen as a model organism of Gram-negative bacteria and staphylococcus as a representative of Gram-positive bacteria for the reason that they have different defence mechanisms at the cellular level against biocides. Disinfectants were selected based on the frequency of use in industry but also on the basis of their chemical composition. Disinfectant-(P) is based on peracetic acid (CH3COOOH, min. 15%), hydrogen peroxide and stabilisers, while disinfectant-(D) represents quaternary ammonium compounds (QACs) based on benzalkonium chloride.

In light of the above, the present study aimed to estimate the efficacy of two disinfectants intended for use in the food industry on planktonic and biofilm populations of *E. coli* and *S. aureus*. In addition, the anti-quorum sensing (anti-QS) activity of the tested disinfectants was investigated.

## 2. Materials and Methods

### 2.1. Bacterial Strain and Preparation of Inoculum

Tests were performed with *S. aureus* ATCC 25923 and *E. coli* isolated from pork meat. Isolation was carried out according to the standard method [21]. Each MO was cultured on tryptone soy agar (TSA, LabM, Lancashire, UK) for 24 h at 37 °C. After incubation, 2 to 3 individual colonies were transferred to 10 mL of tryptone soy broth (TSB, Oxoid, Hampshire, UK). The suspensions were incubated at 37 °C for 18 h. The final concentration of the bacterial suspensions used in the experiments was obtained by adjusting the cell density according to the 1.0 McFarland standard (~3 × 10^8^ CFU/mL) using a DEN-1 densitometer (Biosan, Riga, Latvia). The quorum biosensor bacterial strain *Chromobacterium violaceum* CV026 (a double mini-Tn5 mutant derived from *Chromobacterium violaceum* ATCC 31532) was used for anti-QS disinfectant activity testing [22].

### 2.2. Disinfectants

The following commercial disinfectants were used: disinfectant (P) and disinfectant (D), both manufactured by Albus (Novi Sad, Serbia). Disinfectant (P) contains peracetic acid (CH3CO3H, min. 15%), hydrogen peroxide and stabilisers. The concentration recommended by the manufacturer for rapid disinfection is 0.3% for 5 to 10 min. Disinfectant (D) is a benzalkonium chloride-based product for disinfecting surfaces that come into contact with food and for all other surfaces that do not come into direct contact with food. The disinfectant contains benzyl C12 C16 alkyl dimethyl chloride, 25% (*w*/*w*). The concentration recommended by the manufacturer for rapid disinfection is 0.3–0.5% for 5 to 10 min.

### 2.3. Examination of Disinfectant Properties against Planktonic Forms

The test of the efficacy of the selected disinfectants was carried out following the German Society for Hygiene and Microbiology (DGHM) with slight modifications [23]. The test applied is based on performing cultivation after contact with the disinfectant on a solid nutrient medium to determine the number of surviving cells. The results obtained were compared with an (untreated) control sample (test organism inoculated into the broth without the addition of a disinfectant).

The test disinfectants (P or D) were inoculated directly into a corresponding bacterial suspension of a specific test MO corresponding to the 1.0 McFarland standard (~3 × 10^8^ CFU/mL) to achieve the targeted disinfectant concentrations (0.1%, 0.3% and 0.5%, respectively). After exposing the bacterial suspensions for 5 and 10 min, the effect of the disinfectants on the planktonic population of MOs was determined using the standard colony counting technique on TSA with the result expressed as log CFU/mL. The number of viable cells was determined in triplicate for each MO tested. To neutralise the further effect of the disinfectant, lecithin (3 g/L) and polysorbate 80 (30 g/L) were added to the TSA.

The germicidal effect (*GE*) of the disinfectants was determined on the basis of the decimal reduction degree according to Equation (1):*GE* = log (*Nc*) − log (*Nd)*,(1)
where *Nc*—number of bacteria without treatment (CFU), and *Nd*—number of bacteria after disinfection treatment (CFU)

### 2.4. Biofilm Formation Assay

The biofilm formation test was based on the previously described method [24,25]. Specifically, 8 wells of sterile, flat-bottomed 96-well polystyrene microtitre plates (Greiner Bio-One GmbH, Frickenhausen, Germany) were filled with 180 µL TSB. Aliquots of 20 µL of each bacterial suspension were then added to each well. The negative control contained only the broth tested, 200 µL per well. The plates were incubated at 25 °C for 48 h. After incubation, the contents of each well were removed, and the wells were washed 3 times with 250 µL of sterile water. Each plate was air-dried in an inverted position at room temperature for 30 min. Subsequently, each well was stained with 250 μL of 0.5% crystal violet (CV) (Sigma-Aldrich, Burlington, MA, USA) for 20 min. Excess stain was rinsed off by filling the wells with sterile water. The microtitre plates were air-dried in an inverted position at room temperature for one hour. The bound CV was dissolved with 250 μL decolouriser (33% acetic acid). The plates were left at room temperature for 15 min to achieve complete colour dissolution. The absorbance (A)* of the wells was measured at 630 nm using an automated microplate reader (Thermo Fisher Scientific, Waltham, MA, USA). Based on the A values obtained, the isolates were classified into four categories [26] with the Ac limit defined using the A values of the negative control (pure broth). Absorbance cut-off value (Ac) = average A of negative control + 3 × standard deviation (SD) of negative control (Table 1).

### 2.5. Examination of Disinfectants Properties against Microbial Cells in Biofilms 

Stainless steel coupons (diameter 1.1284 cm) were used for biofilm formation. The coupons were previously washed in a rinsing agent solution and rinsed with sterile distilled water. They were then sterilised in an autoclave (Tuttnauer, ELV 3870, Bet Shemesh, Israel) at a temperature of 121 °C for 15 min. Each coupon was placed individually in the wells of a sterile 12-well polystyrene plate (Greiner Bio-One, Frickenhausen, Germany). The suspension of test bacterial isolates was inoculated in 100 µL onto the surface of each coupon. Adhesion of the bacteria was ensured during incubation at 25 °C for 48 h. During adhesion, the surface of each disc was inoculated with 100 µL of a bacterial suspension after 12 h and after 24 h. The bacteria were inoculated onto the surface of each disc. The discs prepared in this way were used for further treatment with the disinfectants tested (Figure 1).

Subsequently, each coupon placed in the plate wells was treated with concentrations of the tested disinfectants (0.1%, 0.3%, 0.5%, 1% and 2%, respectively) at exposure times of 5 min, 10 min and 20 min. Afterwards, the coupons were washed 3 times with 1 mL sterile distilled water and then placed in tubes containing 1 mL saline (Himedia, Mumbai, India). Finally, the bacteria were detached from the coupons by exposing the tubes with coupons to low-energy ultrasound at a frequency of 40 kHz for 3 min using an ultrasonic water bath (Ultrasonic Cleaner, Electric VIMS, Tršić, Serbia) and centrifugation at 8000 rpm for 5 min (Eppendorf, Hamburg, Germany).

The tubes were shaken at maximum speed (40 Hz) for 1 min, and the bacteria were resuspended in 9 mL peptone saline (Himedia, India). The number of adherent cells was determined using a standard colony-counting procedure on TSA. Plates were incubated at 37 °C for 24 h. Three coupons were analysed for each strain tested, and the results were expressed as log CFU/cm^2^.

To neutralise the further effect of the disinfectant, lecithin (3 g/L) and polysorbate 80 (30 g/L) were added to the TSA. The germicidal effect of the disinfectants (GE) on the biofilms formed was determined on the basis of the decimal log reduction ratio according to the above Equation (1).

### 2.6. Anti-Quorum Sensing Activity

The anti-QS potential of the tested disinfectants was assayed using *C. violaceum* CV026 as a biomonitor strain according to the previously described disc diffusion method [27]. The test was conducted on the Luria–Bertani agar (LBA, Himedia, Mumbai, India) supplemented with HHL (N-hexanoyl-DL-homoserine lactone) 10 μL/50 mL LBA medium. *C. violaceum* CV 026 was grown overnight in Luria–Bertani broth at 30 ± 3 °C. After that, 100 μL of bacterial suspension (≈10^8^ CFU/mL) was poured over LB agar plates supplemented with HHL. Subsequently, sterile discs (6 mm diameter) were placed over the agar plates and loaded with 20 µL of tested disinfectant in respective concentrations: 0.3%, 0.5% and 1%. Antimicrobial (clear ring) and quorum sensing inhibition (a ring of colourless but viable cells) were measured after 24 h of incubation at 30 ± 3 °C.

### 2.7. Statistical Analysis

Three operational parameters (*X*_1_—initial number of MOs, *X*_2_—disinfectant concentration and *X*_3_—time) were independent factors in the chosen experimental design. The germicidal effect of disinfectant P or D on the reduction of pre-formed *E. coli* and *S. aureus* biofilms was chosen as the dependent factor (*Y_k_*). The response surface method was used to evaluate the influence of the operating parameters on the germicidal effect for each bacterial strain. The relationships between the independent factors and the responses were calculated with the second-order polynomial Equation (2):(2)$$Y_{k}=b_{0}+\sum_{i=1}^{3}b_{i}\cdot X_{i}+\sum_{i=1}^{3}X_{i}^{2}+\sum_{i=1,j=i+1}^{3}X_{i}\cdot X_{j},k=4,$$
where *Y_k_* is the defined response; *b*_0_ is the intercept, *b_i_*, *b_ii_* and *b_ij_* are the linear, quadratic and interaction regression coefficients respectively, while *X_i_* and *X_j_* are the varied factors.

Statistical analyses were performed using Statistica v. 13.2 software (Dell, Round Rock, TX, USA). The influence of the factors studied, as well as their interaction, was investigated by comparing the sum of squares values for each of the coefficients in the second-order polynomial model (SOP). Response surface plots were drawn using the same software for a constant value of an initial number of MOs (7.46 log CFU/cm^2^ for *S. aureus* and 7.82 log CFU/cm^2^ for *E. coli*) and varied values of the other two factors (disinfectant concentration and time).

## 3. Results

### 3.1. Influence of Disinfectants on Broth Cultures of Tested MOs

The results of the germicidal effect of the tested disinfectants P and D indicate that all tested strains showed a significant decrease in the survival rate of viable cells after treatment with the disinfectants tested. The germicidal effect of the 2 disinfectants tested was achieved at the lowest concentration used of 0.1% and an exposure time of 5 min for both MOs tested.

### 3.2. Biofilm Formation Ability

The results of the biofilm-forming ability of *E. coli* and *S. aureus* are shown in Figure 2. Both MOs tested formed a biofilm on microtitre plates at a temperature of 25 °C. Based on the results from A, the tested MOs *E. coli* and *S. aureus* were characterised as strong biofilm producers (4 × A_c_).

### 3.3. Influence of Disinfectants on Formed Biofilms on Steel Discs

The germicidal effect (GE) of different concentrations of disinfectants P and D at an exposure time of 5 min, 10 min and 20 min on biofilms of *E. coli* and *S. aureus* formed for 48 h previously is shown in Table 2 and Table 3. According to the results, the disinfectants used had a significantly weaker GE on the bacterial populations studied in the biofilms when compared to the exposure of broth cultures of the same MOs with the same disinfectant concentrations.

When observing the effect of disinfectants P (Table 2) and D (Table 3) in a concentration of 0.1% on the biofilm of *S. aureus* formed at 48 h, based on the obtained results, it can be concluded that when the biofilm is exposed to this concentration for 5 and 10 min, disinfectant D has a better germicidal effect. A similar conclusion can be drawn when it comes to the 48 h biofilm of *E. coli* if the same concentration of disinfectants P and D is observed for the same time of action.

At a higher concentration (0.3%) of tested disinfectants P (Table 2) and D (Table 3), an additional decrease in the number of viable cells of *S. aureus* and *E. coli* biofilms can be observed compared to concentrations of 0.1%. It can also be concluded that disinfectants P and D at a concentration of 0.3% for the same time of action had a greater degree of reduction in the number of viable cells when it comes to the *E. coli* biofilm compared to the reduction of viable cells of the *S. aureus* biofilm. The lowest GE of disinfectant P on the *S. aureus* biofilm was found at the lowest tested concentration of 0.1% with the shortest exposure time (5 min). For *E. coli*, a higher GE was achieved under the same experimental conditions (Table 2). A similar effect was also found for disinfectant D (Table 3). With increasing disinfectant concentration and exposure time, GE increased for both MOs tested. Complete destruction of viable cells in biofilms of the 2 MOs tested (*S. aureus* and *E. coli*) was achieved after 5 min of exposure time of the highest concentration (2%) of each disinfectant (P or D).

The obtained results shown in Table 2 and Table 3 indicate that the disinfectants in the applied concentrations cause a reduction in the number of viable cells in the biofilms in a dose-dependent manner as a function of time. 

ANOVA for the obtained second-order polynomial models (SOP) for the germicidal effect of disinfectant P or D on the reduction of pre-formed biofilms of *E. coli* and *S. aureus* was performed, and the response variables were tested for the influence of the factor variables (Table 4).

According to the results, the germicidal effect of disinfectant P on the reduction of pre-formed biofilms of *S. aureus* was mainly influenced by the quadratic term of disinfectant concentration and the nonlinear exchangeable term between the initial number of MOs and disinfectant concentration (statistically significant at *p* < 0.05).

Similarly, the germicidal effect of disinfectant D on the reduction of pre-formed *S. aureus* biofilms was mainly influenced by the quadratic term of disinfectant concentration and the interaction term between the initial number of MOs and disinfectant concentration (statistically significant at *p* < 0.05). The germicidal effect of disinfectant D on the reduction of *S. aureus* biofilms was also influenced by the linear term of time (statistically significant at *p* < 0.05). The coefficients of determination (r^2^) for the models of SOP were quite good (0.925–0.968) (Table 4). According to the results in Table 4, the higher r^2^ values were attributed to the models from SOP in which the nonlinear terms had a greater effect and were more pronounced. The relatively inaccurate results of the models from SOP suggest that perhaps other nonlinear models would improve the validity of the model predictions.

Three-dimensional response (RSM) surface plots were created to show the interactions of the operational parameters and to define the recommendation of their value in the germicidal effect of disinfectant P or D on the reduction of pre-formed biofilms of *E. coli* and *S. aureus*. All RSM plots show the interaction of two tested parameters on the germicidal effect of disinfectant P or D on the reduction of pre-formed biofilms of *E. coli* and *S. aureus*. In contrast, the third parameter was kept at the central value of the experimental design (the initial number of MOs was set at 7.46 log CFU/cm^2^ for *S. aureus* and 7.82 log CFU/cm^2^ for *E. coli*).

Based on the results for the germicidal effect of disinfectant P or D on reducing *E. coli* and *S. aureus* biofilms (Figure 3), it is clear that the maximum germicidal effect was achieved at the maximum disinfectant concentration at each time value. A similar effect was observed for each disinfectant P or D and the reduction of all *E. coli* or *S. aureus* biofilms (Figure 3a–d). In the case of disinfectant D (Figure 3c,d), the maximum GE (about 7.91) is observed at the highest values of the two parameters presented. On the other hand, the maximum value of GE for disinfectant P (about 7.77) was obtained at the same parameters (Figure 3a,b).

The results shown in Figure 3a–d suggest that the maximum germicidal effect of disinfectant P or D on the reduction of pre-formed *E. coli* and *S. aureus* biofilms was achieved at the highest values of disinfectant concentration (2%, regardless of disinfectant type, P or D).

### 3.4. Anti-Quorum Sensing Activity of Tested Disinfectants on Chromobacterium violaceum

The test results of the anti-QS activity of the tested disinfectants P and D are shown in Figure 4 and Figure 5. At the tested concentrations of 0.3%, 0.5% and 1%, the inhibition zones of disinfectant D were 18.33 ± 0.5 mm; 23.0 ± 1.0 mm and 28.3 ± 0.5 mm, respectively. At the same concentrations tested, the inhibition zones of disinfectant P were smaller and were 15.0 ± 1.0 mm, 19.0 ± 1.0 mm and 24.6 ± 0.5 mm, respectively.

Considering that anti-QS was detected as a colourless, opaque zone around the disc manifested by bacterial growth without violacein synthesis, it can be concluded on the basis of the test results that the tested disinfectants P and D have no anti-QS effect. The inhibition zones around the disc thus correspond to an antimicrobial effect without an anti-QS effect (Figure 5).

## 4. Discussion

### 4.1. Influence of Disinfectants on Broth Cultures of Tested MOs

The results showed a significant reduction in the survival rate of viable cells of all strains studied after treatment with specific disinfectants. After 10 min of exposure to iodine (0.2%), biguanides (0.5%), quaternary ammonium compounds (0.5%), peracetic acid (0.5%) and sodium hypochlorite (1.5%), no viable planktonic cells were observed. These results correlate well with the data of the present study (Table 2 and Table 3). Similar results were obtained in a previous study on the effect of different disinfectants on planktonic cells of *S. aureus*, *E. coli* and *L. monocytogenes* [28].

In another study [29], the efficacy of Oxsil^®^320N on bacterial suspensions in broth was investigated. This disinfectant contains three active ingredients (hydrogen peroxide, acetic/peracetic acid and silver) with different mechanisms of action. In this study, a GE was achieved after 5 min of exposure at 0.031% for *E. coli* and *S. aureus*, 0.039% for *Pseudomonas aeruginosa* and 0.313% for *Enterococcus hirae*. The efficacy of a very low concentration of Oxsil^®^320N on *E. coli* and *S. aureus* can be explained by the synergism of several active ingredients contained in this disinfectant.

### 4.2. Biofilm Formation Ability

The higher adhesion capacity of Gram-negative bacteria has been described previously [30,31]. Another study showed that *S. aureus* cells adhered in lower numbers (despite the higher initial concentration of cells in the suspension used for subsequent biofilm formation), while the highest adhesion capacity was shown by *Pseudomonas fluorescens* followed by *E. coli* [32]. Evaluation of adhesion to wood surfaces revealed that the Gram-negative bacteria *E. coli* ATCC 35218 and *P. aeruginosa* ATCC 27853 showed significantly better adhesion ability than the Gram-positive bacterium *S. aureus* ATCC 25923 [33]. These data are consistent with our observations in this study (data in Figure 2).

### 4.3. Influence of Disinfectants on Formed Biofilms

In some cases, the weaker effect of the disinfectants on the pre-formed biofilms could be explained by the presence of the extracellular matrix (ECM), which could act as a diffusion barrier to varying degrees. It has already been mentioned that the ECM is a barrier to antimicrobial substances and prevents their transport to the bacteria primarily by interacting with antimicrobial substances and inactivating them or producing enzymes that degrade them. The ECM surrounding the biofilm acts as a diffusion barrier, molecular sieve or adsorbent to varying degrees. Diffusion of the antimicrobial substance through the ECM towards the deeper layers reduces its concentration. Thus, only the bacteria in the biofilm’s surface layers are exposed to lethal concentrations. By slowing down the penetration of antimicrobial substances, the ECM of the biofilm also acts as a dilution gradient, giving the cells additional time to activate the expression of new resistance genes before they are actually exposed to the antimicrobial substances [15,34]. The results obtained are supported by the fact that certain components of the ECM play a structural role by providing mechanical, chemical and biological protection in the natural habitat. Several studies also indicated that the presence of cellulose in the ECM contributes to greater viability and is directly responsible for resistance to treatment with various antimicrobial agents [12,35,36].

Antimicrobial resistance can also result from mutation, amplification of chromosomal genes or acquisition of resistance determinants from extrachromosomal genetic elements (such as plasmids and transposons) [37]. Resistance arising from disinfectant inactivation is known but is a relatively rare case and specific to several classes of pathogens. Much more commonly and probably due to the existence of multiple cellular targets for antimicrobial agents, resistance is the result of enhanced efflux from the cell or a change in the permeability of the cell membrane. This is confirmed by studies by Sundheim et al. [38] who report that Gram-positive bacteria are generally sensitive to quaternary ammonium compounds (QACs). However, some staphylococci contain genes encoding an efflux system that allow QACs to be pumped out of the cell using a proton-driven transporter (PMF), a transmembrane electrochemical proton gradient.

In addition, some studies have shown that slow growth and the induction of an rpoS-mediated stress response may contribute to antimicrobial resistance. As early as 1999, Adams and McLean [39] reported that the release of rpoS significantly reduced the ability of *E. coli* to grow in the form of a biofilm and had little effect on the growth of planktonic bacteria. The physical and chemical structure of exopolysaccharides or other compounds incorporated into the structure of a biofilm can also provide resistance by keeping disinfectants away from the bacterial community. In addition, bacteria that grow in a biofilm can develop a biofilm-specific phenotype that is resistant to a particular antimicrobial agent.

The antimicrobial effect of disinfectants on MOs also depends on the concentration used. The lower concentrations usually cause increased permeability of the cell membrane, which disrupts its structure without affecting viability. At the same time, higher concentrations lead to greater damage to the membrane, complete disruption of homeostasis and cell death. Accordingly, prolonged contact between disinfectants and MOs can cause severe damage, inevitably leading to bacterial cell death. This mode of action may explain the higher efficiency of disinfectants after a longer exposure time (20 min) compared to treatment with a shorter exposure time (5 min and 10 min). A longer exposure time for bacteria in the biofilm probably leads to a stronger diffusion of the applied substances through the biofilm matrix. This mode of action of disinfectants was illustrated in studies by Solano et al. [35].

The results obtained in this paper are consistent with the findings of other studies, which found that current sanitary practices are less useful for adherent MOs that form biofilms than for free planktonic forms [40,41,42,43,44,45]. Other studies have also found that bacteria in biofilms are able to survive higher concentrations of quaternary ammonium compounds (QACs) than their planktonic forms. Interestingly, they showed that QACs at concentrations below the MIC limit induce biofilm formation in *S. aureus* but can inhibit biofilm formation in *E. coli*. These results suggest that exposure of some bacteria to low concentrations of QACs may stimulate biofilm formation and subsequently lead to higher survival after disinfection [46].

Costa et al. [47] investigated the effect of peracetic acid on biofilm formation. After contaminating stainless steel with *S. aureus*, *E. coli* and *Candida albicans*, they immersed it in peracetic acid at concentrations of 0.25% and 2% for 30 min. Both concentrations tested resulted in a significant elimination of the MOs. However, the effectiveness of the disinfectants in reducing MOs was significantly lower when they were present in the form of a biofilm than in broth cultures.

The efficacy of various disinfectants, such as iodine (0.2%), biguanides (0.5%), quaternary ammonium compounds (0.5%), peracetic acid (0.5%) and sodium hypochlorite (1.5%), on planktonic and cells in the biofilms of *L. monocytogenes*, *S. aureus* and *E. coli* was also studied [28]. The study found a reduction in the number of viable biofilm cells after treatment with any of the disinfectants tested (sodium hypochlorite was the most effective and biguanide least effective). However, scanning electron microscopy showed the presence of adherent cells of these MOs on the surface of stainless steel discs (AISI type 304) even after a 10 min treatment with disinfectants, while the presence of viable planktonic cells could not be detected after the same treatment. Thus, it was obvious that biofilm cells might be more resistant to disinfectants compared to planktonic cells. Furthermore, our results (Table 2, Table 3 and Table 4 and Figure 3) correlate well with previously published studies.

Holah et al. [48] and Meyer and Cookson [49] also evaluated the efficacy of disinfectants to remove biofilm cells and found that efficacy increased from quaternary ammonium compounds to amphoteric compounds, chlorine, biguanides and peroxyacids. Peracetic acid was also found to be more effective in removing adherent *S. aureus* cells compared to hydrogen peroxide and sodium dichloroisocyanurate [50].

After comparing the biocidal concentrations of the disinfectant Oxsil^®^320N for planktonic cells and MOs in biofilms, it was found that for the same exposure time (5 min), the biocidal concentrations for biofilms were significantly higher and were up to 12.52% for *E. hirae* and *S. aureus*, 46.87% for *P. aeruginosa* and 62.5% for *E. coli* [29].

Studies by other authors also showed higher efficacy of different disinfectants on planktonic cultures of *E. coli* and *S. aureus* than on pre-formed biofilms of the same MOs [51,52,53,54,55]. They also concluded that a significantly higher concentration is required to reduce MOs in biofilm form.

Solano et al. [35] suggested that the presence of cellulose in the biofilm matrix of some bacteria not only contributes to increased survivability but is also directly responsible for bacterial tolerance to treatments with various antimicrobial agents. 

Based on the presented results and literature data, it can be concluded that the greater resistance of formed biofilms is largely due to the presence of the biofilm matrix, but the possible influence of other resistance mechanisms of bacteria in a biofilm cannot be excepted.

### 4.4. Anti-Quorum Sensing Activity

In contrast to the present results (Figure 4 and Figure 5), the studies by Venkadesaperumal et al. [27] indicated anti-QS activity of nanoemulsions of cumin (*Cuminum cyminum*), pepper (*Piper nigrum*) and fennel (*Foeniculum vulgare*). The disc diffusion method where they used oil nanoemulsions in an amount of 50 µL showed their anti-QS activity against strains of *C. violaceum* CV026. In addition, the oil nanoemulsions had an inhibitory effect on the synthesis of violacein around the discs, as shown by a loss of purple pigmentation. The nanoemulsions tested showed a clean zone immediately around the disc (indicating a bactericidal effect) accompanied by an opaque halo, indicating inhibition of violacein synthesis. Similar results were obtained for the anti-QS potential of the essential oil and phenols of *Carum copticum* against *C. violaceum* [56]. Asghar et al. [57] found that of the CEO concentrations tested (essential oil of green cardamom—*Elletaria cardamomum*), 0.625 and 0.313 mg/mL were effective in inhibiting bacterial quorum sensing by testing violacein production with minimal effects on *C. violaceum* growth.

## 5. Conclusions

In conclusion, the results obtained in the current study revealed that the test MOs *E. coli* and *S. aureus* were characterised by strong biofilm production ((4 × A_c_) < A). Furthermore, *E. coli* showed significantly higher adherence ability compared to *S. aureus*. The analysis of the effect of disinfectants P (based on peracetic acid) and D (based on benzalkonium chloride) on *E. coli* and *S. aureus* in suspension showed that even at the lowest concentration used (0.1%) and during the shortest exposure time (5 min), a germicidal effect of 100% was achieved. In contrast, complete destruction of viable cells in biofilms was achieved after 5 min treatment with disinfectants P or D but only when using the highest concentration (2%).

Considering that biofilms are common in the food industry and affect food safety, it is of utmost importance to continuously test the efficacy of different antimicrobial agents that would penetrate the complex structures of biofilms with the lowest possible concentration in the shortest possible time and exert an antimicrobial effect.

## Figures and Tables

**Figure 1 microorganisms-11-01593-f001:**
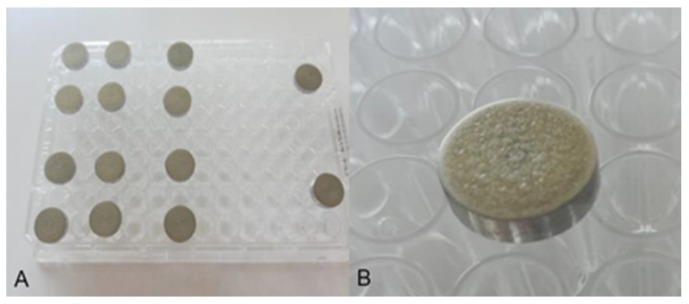
Stainless steel coupons (AISI 304) after the biofilm formation (**A**) and enlarged (**B**).

**Figure 2 microorganisms-11-01593-f002:**
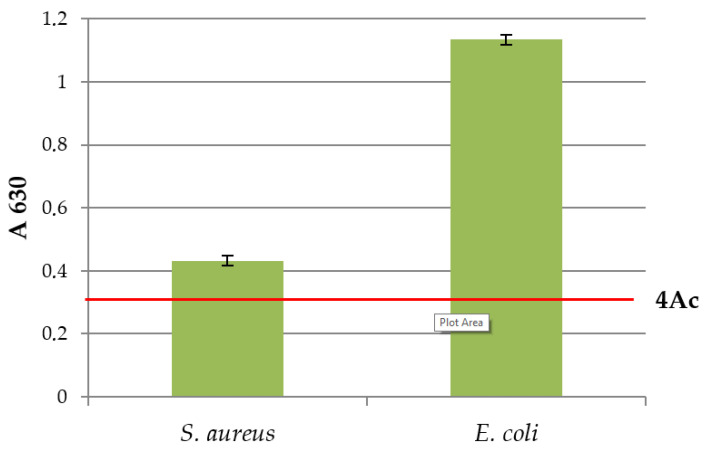
Biofilm forming ability of *E. coli* and *S. aureus* at 25 °C.

**Figure 3 microorganisms-11-01593-f003:**
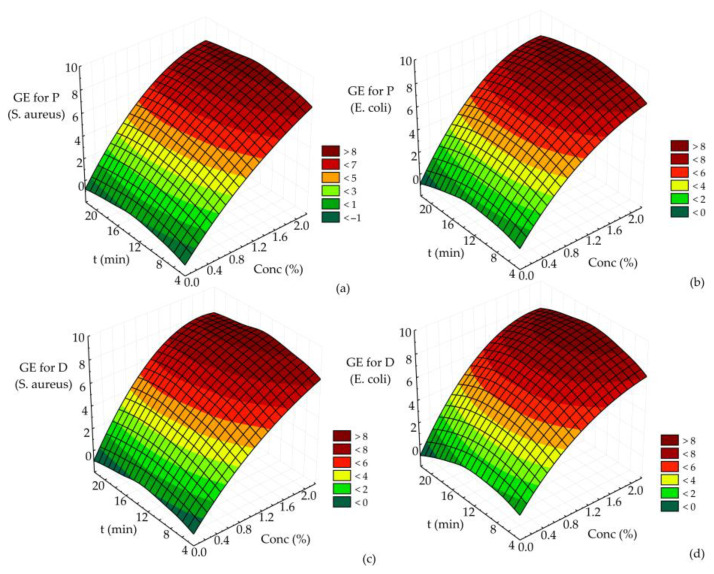
The germicidal effect of (**a**) disinfectant P on the reduction of pre-formed *S. aureus* biofilms, (**b**) disinfectant P on the reduction of pre-formed *E. coli* biofilms, (**c**) disinfectant D on the reduction of pre-formed *S. aureus* biofilms and (**d**) disinfectant D on the reduction of pre-formed *E. coli* biofilms.

**Figure 4 microorganisms-11-01593-f004:**
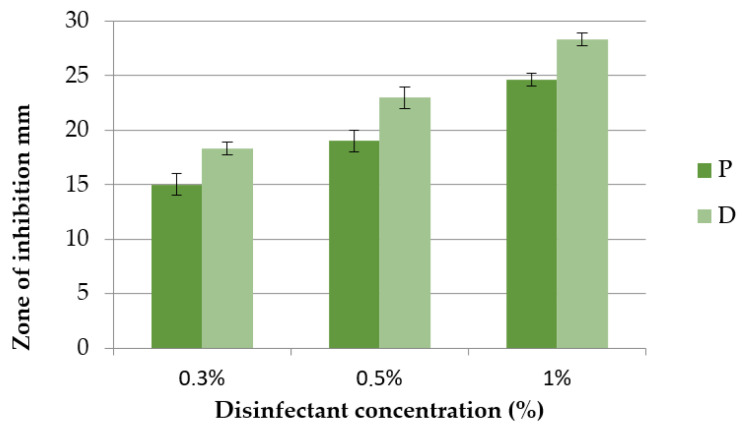
Zone of inhibition of *C. violaceum* after exposure to disinfectants P and D at concentrations of 0.3%, 0.5% and 1%.

**Figure 5 microorganisms-11-01593-f005:**
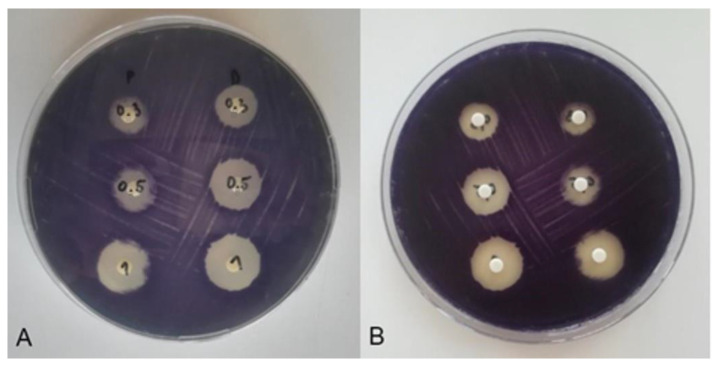
Inhibition zones of *C. violaceum* after exposure to P and D disinfectants at concentrations of 0.3% (first row of discs), 0.5% (second row of discs) and 1% (third row of discs). (**A**) Back view of the Petri dish: disinfectant P (first column of discs) and disinfectant D (second column of discs). (**B**) Display of the Petri dish so that the discs are in the foreground: disinfectant D (first disc column) and disinfectant P (second disc column).

**Table 1 microorganisms-11-01593-t001:** Representation of isolate categorisation based on absorbance.

Absorbance (A)	Biofilm Production
A ≤ A_c_	there is no biofilm production
A_c_ ≤ OD ≤ (2 × A_c_)	weak biofilm production
(2 × A_c_) < A ≤ (4 × A_c_)	moderate biofilm production
(4 × A_c_) < A	strong biofilm production

A—absorbance of the sample at 630 nm; A_c_—limit value.

**Table 2 microorganisms-11-01593-t002:** Germicidal effect of disinfectant P on the reduction of pre-formed biofilms of *E. coli* and *S. aureus* biofilms.

Tested MO	The Initial Number of MOs (log CFU/cm^2^)	Ambient Temperature (°C)	Disinfectant Concentration (%)	GE_5min_	GE_10min_	GE_20min_
*S. aureus*	7.19	23	0.1	0.15	0.65	-
	7.19	23	0.3	0.32	1.24	-
	7.19	23	0.5	2.05	3.63	4.46
	7.77	23	1.0	3.34	4.26	5.69
	7.77	23	2.0	7.77	7.77	7.77
*E. coli*	7.73	23	0.1	0.63	1.11	-
	7.73	23	0.3	1.45	2.99	-
	7.73	23	0.5	3.50	4.75	5.55
	7.91	23	1.0	4.08	4.54	5.73
	7.91	23	2.0	7.91	7.91	7.91

The germicidal effect of disinfectant P in concentrations of 0.1% and 0.3% on the reduction of 48 h biofilms of *S. aureus* and *E. coli* has not been studied.

**Table 3 microorganisms-11-01593-t003:** Germicidal effect of disinfectant D on the reduction of pre-formed biofilms of *E. coli* and *S. aureus* biofilms.

Tested MO	The Initial Number of MOs (log CFU/cm^2^)	Ambient Temperature (°C)	Disinfectant Concentration (%)	GE_5min_	GE_10min_	GE_20min_
*S. aureus*	7.19	23	0.1	0.20	0.96	-
	7.19	23	0.3	0.46	1.50	-
	7.19	23	0.5	2.20	4.03	4.81
	7.77	23	1.0	3.50	4.31	6.62
	7.77	23	2.0	7.77	7.77	7.77
*E. coli*	7.73	23	0.1	0.75	1.64	-
	7.73	23	0.3	1.97	3.58	-
	7.73	23	0.5	3.76	5.04	5.75
	7.91	23	1.0	4.22	4.79	5.95
	7.91	23	2.0	7.91	7.91	7.91

The germicidal effect of disinfectant P at concentrations of 0.1% and 0.3% on the reduction of 48 h biofilms of *S. aureus* and *E. coli* was not studied.

**Table 4 microorganisms-11-01593-t004:** ANOVA calculation of the germicidal effect of disinfectant P or D on the reduction of pre-formed biofilms of *E. coli* and *S. aureus* (sum of squares is shown).

Factor	df	P *S. aureus*	P *E. coli*	D *S. aureus*	D *E. coli*
*INM*	1	2.916	1.583	3.422	1.135
*Conc*	1	1.468	0.674	1.741	0.405
*Conc* ^2^	1	3.397 **	2.464	3.809 **	1.895
*t*	1	2.346	0.500	3.961 **	0.349
*t* ^2^	1	0.797	1.308	0.972	2.074
*INM* × *Conc*	1	3.784 **	2.995	4.298 **	2.385
*INM* × *t*	1	0.653	0.444	1.677	0.630
*Conc* × *t*	1	0.387	0.077	0.945	0.078
Error	6	3.926	6.476	5.707	7.999
R^2^		0.968	0.942	0.954	0.925
adj R^2^		0.925	0.863	0.892	0.825

*INM*—initial number of MOs; *Conc*—disinfectant concentration; *t*—time; P *S. aureus*—effect of disinfectant P on reduction of pre-formed biofilms of *S. aureus*; P *E. coli*—disinfectant P on reduction of pre-formed biofilms of *E. coli*; D *S. aureus*—disinfectant D on reduction of pre-formed biofilms of *S. aureus*; D *E. coli*—disinfectant D on reduction of pre-formed biofilms of *E. coli*. **—Statistically significant at *p* < 0.05 level.

## Data Availability

Data available upon request.

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
