# Peer review of "Comparison of the Efficiency of Selected Disinfectants against Planktonic and Biofilm Populations of Escherichia coli and Staphylococcus aureus"

_microorganisms, 2023, doi:10.3390/microorganisms11061593_

Round 1
Reviewer 1 Report
It is very unfortunate that the manuscript is not well organised. The rationale for choosing the two disinfectants and the two bacterial species to test is not explained. The authors did not state why this study is novel. The statistical analysis in this study is concerning.
Detailed comments:
Introduction, the rationale for choosing the two disinfectants and the two bacterial species to test should be explained.
Line 77, please correct “Tripton”.
Line 78, one colony is enough. Why were two to three colonies picked?
Line 130-135, it should be absorbance instead of optical density, please correct it in these sentences and in table Table 1.
Line 133, the definition of ODc is not clear.
Line 142-147, the description is not clear. Did the authors mean cultures were added onto the coupon for three times? In Figure 1, why the coupons are round? Line 138 mentions its dimension, indicating it is square. Please explain.
Line 159, how was the peptone saline made?
Tables 2 and 3 are redundant and should be removed. The results shown in these two tables can be summarised using one sentence.
Tables 4 and 5, why the authors did not do the test for the “-“ parts? Please explain.
Lines 239-248, and 262, why was 0.1 used for P value threshold instead of 0.05?
Line 239-248, please interpretate the results using biological rather than statistical terms.
Lines 266 and 276, which two terms?
Also, in the results section, the two disinfectants should be compared.
Discussion, this part should be re-organised. Instead of discussing too much about other studies, the authors should focus on discussing the present study.
The reviewer strongly suggest the authors to go through the whole manuscript to check the English.
Reviewer 2 Report
This manuscript was well written and prepared, and for biofilm formation potential by different disinfectants, it is useful for food safety practice. 1) Why to focus on these two strains? Actually, these two strains were as the bacterial indicator, but in specific foods they would be specific spoilage or pathogens, maybe specific bacteria affected by these infectants would be more useful. 2) These infectants, P and D should be added for more explanation for differences.
Round 2
Reviewer 1 Report
The manuscript has been largely improved. There are still a number of issues that should be addressed before it is acceptable for publication.
1. The authors have explained the rationale for choosing two disinfectants and the two bacterial species in the response to the reviewer. Please also add the relevant statements should be added in the manuscript as well.
2. Line 131, Yes, we can find the use of OD in biofilm CV staining method in some papers, it does not mean OD is the right term.
Please see the interpretation about the different between optical density and absorbance. https://www.differencebetween.com/difference-between-optical-density-and-absorbance/#:~:text=Optical%20density%20is%20the%20degree,light%20of%20a%20specified%20wavelength.&text=Measurement-,The%20optical%20density%20measurement%20takes%20both%2C%20the%20absorption,scattering%20of%20light%2C%20into%20consideration.
This is why we use optical density to describe the biomass of bacterial growth. For the case in this study, the absorbance of CV was determined.
3. Regarding the statistical analysis, normally P < 0.05 is used. We do not regard P=0.05 as significant.
Please go through the language in the whole manuscript to make it easy to understand for the audience.
